# Nanotechnology and E-Sensing for Food Chain Quality and Safety

**DOI:** 10.3390/s23208429

**Published:** 2023-10-12

**Authors:** Elisabetta Poeta, Aris Liboà, Simone Mistrali, Estefanía Núñez-Carmona, Veronica Sberveglieri

**Affiliations:** 1Department of Life Sciences, University of Modena and Reggio Emilia, Via J.F. Kennedy, 17/i, 42124 Reggio Emilia, RE, Italy; 2Department of Chemistry, Life Science and Environmental Sustainability, University of Parma, Parco Area delle Scienze, 11/a, 43124 Parma, PR, Italy; aris.liboa@unipr.it; 3Nano Sensor System srl (NASYS), Via Alfonso Catalani, 9, 42124 Reggio Emilia, RE, Italy; simone.mistrali@nasys.it; 4National Research Council, Institute of Bioscience and Bioresources (CNR-IBBR), Via J.F. Kennedy, 17/i, 42124 Reggio Emilia, RE, Italy; estefania.nunezcarmona@ibbr.cnr.it

**Keywords:** food nanotechnology, e-sensing, electronic nose, electronic tongue, food quality, food safety, food sensor

## Abstract

Nowadays, it is well known that sensors have an enormous impact on our life, using streams of data to make life-changing decisions. Every single aspect of our day is monitored via thousands of sensors, and the benefits we can obtain are enormous. With the increasing demand for food quality, food safety has become one of the main focuses of our society. However, fresh foods are subject to spoilage due to the action of microorganisms, enzymes, and oxidation during storage. Nanotechnology can be applied in the food industry to support packaged products and extend their shelf life. Chemical composition and sensory attributes are quality markers which require innovative assessment methods, as existing ones are rather difficult to implement, labour-intensive, and expensive. E-sensing devices, such as vision systems, electronic noses, and electronic tongues, overcome many of these drawbacks. Nanotechnology holds great promise to provide benefits not just within food products but also around food products. In fact, nanotechnology introduces new chances for innovation in the food industry at immense speed. This review describes the food application fields of nanotechnologies; in particular, metal oxide sensors (MOS) will be presented.

## 1. Introduction

Nanoscience and nanotechnology are the new frontiers of this century [1]. Their applications to the agriculture and food sectors are relatively recent compared with their use in drug delivery and pharmaceuticals [2]. The word ‘nano’ comes from the Greek ‘nanos’, meaning ‘dwarf’ [3]. Nanotechnology is a branch of applied science and technology that deals with the control of matter at the nanometre scale, a billionth of a meter, and the design and implementation of devices at that scale [4]. Nanotechnology used in the agri-food sector ranges from the use of e-sensing devices to detect specific molecules, to nanoparticles to improve the characteristics of food packaging [5]. In fact, organic gasses are emitted from a variety of sources and some of them have biological, chemical, or physical significance in practical settings, which makes gas-sensing, in particular volatile organic compound (VOC) sensing, a growing and highly valued field [6]. For example, VOCs are prevalent in urbanized areas as they are released from industrial processes, transportation activities, residences, or natural resources [7,8]. Therefore, monitoring the level of these vapours is imperative to prevent them from exceeding safety limits, as long-term exposure to certain VOCs has been proven to raise the risk of serious diseases such as cancer and respiratory issues [9]. Furthermore, VOCs can also be utilized as biomarkers for the diagnosis and monitoring of different diseases [10]. For example, acetone and ethanol are very often found in the exhaled breath of diabetics, pentane is abundant in the breath of schizophrenia patients, and high aldehyde concentrations are present in the breath of patients with lung cancer [11]. The detection of such volatile compounds can be performed via electronic noses (e-noses), which are composed of cross-reactive sensor arrays that can interact with a wide variety of VOCs and produce multidimensional outputs.

As with nanosensors, nanoparticles also have a special place in nanoscience and nanotechnology, not only because of their properties resulting from their reduced dimensions, but also because they are promising building blocks for more complex nanostructures [12]. Different nanoparticles such as silver, copper, or chitosan and metal oxide nanomaterials such as titanium oxide or zinc oxide have been reported to have antibacterial properties [13]. Thus, the use of nanotechnology in the food packaging sector helps extend the shelf life of food products by providing a suitable environment [14] which is responsible for minimizing the risk of microbial degradation, moisture loss, gas transmission, and losses of the organoleptic characteristics of food and food products by controlling enzymatic and metabolic activities [15]. Hence, nanotechnology can be used from the beginning to the end of the food chain [16] in various fields such as agriculture (to optimize harvests or identify any plant diseases via nanosensors), transport and storage (development of new packaging materials via nanoparticles), and production of meat products and dairy products (control animal health during breeding via nanosensors).

In this scenario, a comprehensive review is important to define the state of the art regarding nanotechnologies’ possible applications in the food industry and give the necessary tools for food business operators to improve their business. As a matter of fact, these industries are often related to traditional, expensive, and inefficient processes which create a great deal of waste. The application of new technologies in such a vast environment can improve productivity, ensuring an overall improvement in food quality, efficiency, and wellness for humans, animals, and the environment.

In other words, the aim of the present review is to cover the potential aspects of nanoscience and its trend-setting appliances in modern agriculture and food production. This review focuses on the use of e-sensing to improve food production, guarantee a high level of microbiological safety, and reduce enzyme activity and oxidation during storage. Nanotechnology applications in the food industry can be utilized to support protected products and extend food shelf life. In fact, nanotechnology introduces new chances for innovation in the food industry at immense speed. This review describes the food application fields of nanotechnology. Finally, we believe that food nanotechnology is extremely important for food safety.

## 2. Chemical Gas Sensors and E-Sensing

The performance improvements achieved over time have allowed for the expansion of the applications of artificial gas-sensing systems for many different purposes [17].

E-sensing systems are an ideal quality control approach that should allow for an accurate and rapid compound determination with little or no sample pretreatment and without the use of reagents [18]. Visual systems (VSs), electronic nose (e-noses), and electronic tongues (e-tongues), designed to mimic the human senses, can be useful for real-time evaluation of fresh food, allowing for sample pre-treatment reduction [19].

With regard to VSs, colorimeters are traditionally applied to objectively define the colour of food, while the use of VSs via computer or mobile phones has rapidly emerged in recent years; the e-nose and e-tongue use specific sensor arrays that can be applied to analyse volatile compounds or liquid samples [20].

Other types of semiconductor sensors operate as follows: On a solid substrate a layer of active material that interacts with the gaseous/chemical compound (or a mixture of chemical compounds) is deposited. The analyte molecules interact with the sensitive layer, causing the change to its physical properties. These variations are ‘transduced’ into an electrical signal, which is used to identify the presence and the concentration of gaseous or volatile compounds (Figure 1) [21].

The e-nose is the most versatile technology among all three electronic systems and comprises methods to distinguish the application of different preservation methods (chilling vs. freezing, for instance), processing conditions (especially temperature and time), adulteration (meat from different species), and shelf life [22].

In this respect, studies have been conducted to assess the freshness of food products such as poultry meat using electronic nose technology and ultra-fast gas chromatography [23].

With both instruments, it has been noted that there are changes in the volatile fraction of the sample, mainly due to bacterial activity. Certainly, traditional methods allow for fast and reliable prediction of the shelf life of the product. However, they have several limitations, including high costs and long times for a single analysis. When cost is an issue, a dedicated electronic nose equipped with an array of chemical sensors can possibly be used. Chemical sensors do not have the sensitivity of the detectors used in gas chromatography; however, they can be used to obtain reliable results when used in conjunction with adequate chemometric analysis of their response signals. In this work, an e-nose prototype with 6 metal oxide semiconductor sensors (MOSs) and two photo-ionization detectors (PID) was used to determine the chicken’s shelf life, and principal component analysis (PCA) was employed to reduce the data dimensionality [24], allowing the researchers to see defined clusters based on the time spent. This means that although this method may not possess the precision required to accurately categorize the samples into specific refrigeration days, it can be precise enough to determine whether a consumer would deem the sample acceptable or not [23]. The e-tongue is another relevant technology to determine adulteration and processing conditions and to monitor shelf life. Finally, the e-eye may provide accurate measuring of colour evaluation and grade marbling levels in fresh products. However, advances are necessary to obtain information that is more closely related to industrial conditions [25]. The characteristics that an ideal sensor should have are the following:High sensitivity to the analyte;High selectivity (low noise) and stability;Low sensitivity to humidity and temperature;High reproducibility and reliability;Short reaction and recovery times;High strength, durability, and ease of calibration and small size [22].

Traditional methodologies (microbiological, chemical, physical, and sensory) show good precision, accuracy, and reliability; however, they are destructive and time-consuming and require expensive equipment [25]. In this regard, e-sensing systems are very efficient devices and can overcome many of these drawbacks. They have some particular characteristics such as rapidity, simplicity, objectivity, and versatility. Therefore, they are potentially useful for at-line or on-line applications.

For food producers and consumers, the e-nose and e-tongue play a crucial role in food quality and safety in food production, food supervision, and daily life [26]. Hence, the sensing techniques are well suited to replace conventional detection systems.

### 2.1. E-Sensing

#### 2.1.1. Electronic Tongue

E-tongues have emerged as rapid and easy-to-use tools for the evaluation of food quality [27]. The e-tongue is an instrument that measures and compares tastes [28,29]. It can have better sensitivity than the human tongue and can detect substances undetectable by their natural counterparts [30]. This device can be used for the recognition, classification, and quantitative determination of multiple component concentrations [31]. In fact, the e-tongue can be thought of as analogous to both olfaction and taste, and it can be used for the detection of all types of dissolved compounds, including volatile compounds which produce odours after evaporation [32]. Liquid samples are directly analysed without any preparation, whereas solids require a preliminary dissolution before measurement [33].

This technology comprises a large range of electrochemical sensors for pattern recognition, capable of recognizing simple and complex odours [34].

Regarding electromechanical systems, they consist of a chamber or array of gas sensors and a pneumatic circuit that conducts the air, including volatiles, from the product to be analysed to the sensor chamber [35,36,37]. The sensors generate an electrical signal in the presence of certain molecular chains (Figure 2) [38].

Nevertheless, a disadvantage of these systems is the huge number of preliminary measurements required for the modelling, calibration, or learning steps [39]. These are essential steps for e-tongue performance that cannot be neglected and must be carefully performed. Improved methods for the assessment of food freshness and shelf life time have been high on the priority list of food manufacturers for many years. In this field, e-tongues have proved their potential in predicting the freshness or spoilage of different food raw materials and products.

#### 2.1.2. Electronic Nose

There are many types of sensors that have been developed to detect specific gasses and vapours since the 1970s, but nowadays the human nose is still the primary ‘instrument’ used to assess the smell or flavour of various industrial products, despite considerable and sustained attempts to develop new electronic instrumentation capable of mimicking its remarkable ability [40].

Similar to electronic tongues, the e-nose is an electromechanical system composed of an array of gas sensors and a pneumatic circuit that conducts the air, including volatiles, from the product to be analysed to the sensor chamber (Figure 3). This instrument allows for the detection of simple odours and complex ones via an array of non-specific or partially specific gas sensors that mimic human olfactory perception [40].

Basically, this device consists of three parts:(1)Sampling system for handling samples during analysis(2)Detection system composed of the sensor array(3)Computer for the acquisition and processing of data [41].

Thanks to particular algorithms, an electronic nose system can build an olfactory map that allows for both qualitative and quantitative analysis; for example, it is capable of detecting the origin of a food product from its smell alone. Most of these algorithms work by comparing a new sample with one previously evaluated. Therefore, they require a tuning period (learning phase) to be carried out prior to the operational phase. At the end of the learning phase, the nose will be able to correctly classify most of the samples that will be submitted [39].

Gas sensors represent the heart of the system, and they can be classified into several types. A few technologies are applied in commercial e-noses:metal oxide semiconductor sensors (MOSs)metal oxide semiconductor field-effect transistor sensors (MOSFETs)conducting polymers (CPs)acoustic sensors: surface acoustic wave (SAW) and bulk acoustic wave (BAW) sensors. The most common bulk acoustic wave sensors are quartz microbalance (QMB) sensors [42].

In recent years, e-nose devices have received considerable attention for their potentialities. They have been applied in various fields, such as environmental monitoring [43,44,45,46,47], health [48,49,50,51,52], and food monitoring, with excellent results [53,54,55]. Commercial e-noses are typically handy and compact [56].

Regarding food applications, an electronic nose could be applied in the detection of microorganisms in tomato sauce [57] and of different moulds in coffee [58], the determination of the shelf life of milk [59], the detection of additives in fruit juices [60], the identification of fruit [61], and discrimination between cheese varieties [62,63]. These few examples show how e-noses have the potential to be used in different ways to assess food quality and identity.

An example of an e-nose is the small sensor system (S3) (Figure 4) [64].

The small sensor system (S3) is a device that has amply demonstrated the advantages of the application of this technology in recent years. The S3 is based on semiconductor metal oxide (MOX) gas sensors [65,66], used with considerable success in several sectors, including food safety [67], quality control [68,69,70], environmental monitoring, and human health, in particular thanks to its high sensitivity, fast responses, and low cost [65]. The mechanism of operation of metal oxide sensors is based on the variation of their electrical conductance caused by interaction with the gaseous and volatile organic compounds [71]. The reaction between the oxygen species adsorbed on the surface of the sensitive element from the environment and the analyte molecules cause a release of electrons, which in turn modulate electrical properties (for example, including electrical conductance or resistance) [72]. The surface roughness of the thin film could be an advantage, since it provides a high surface-to-volume ratio and reactivity with gaseous species [73]. In addition, the existence of such rough surface morphology gives rise to the highly specific area required for high-sensitivity gas sensors [74].

In other words, the change is due to a reversible reaction between the volatile compounds and the oxygen species adsorbed on the surface of the semiconductor (SnO_2_, WO_3_, ZnO, CuO, or NiO) [75,76] as O_2_^−^ and especially O^−^ (Figure 5). The superoxide ion, once adsorbed on the surface of the sensor, generates a space charge region and a potential barrier at the edges of the sensor that is opposed to the electric conduction; for this reason, when the sensor is immersed in air, the oxygen concentration is high and the material is characterized by a high resistance (maximum).

### 2.2. Metal Oxide Gas Sensors: MOSs

Conductometric gas sensors based on semiconducting metal oxides are among the most promising solid-state gas sensors thanks to their unique characteristics [77]:-High sensitivity to a broad range of chemicals.-Reduced size and weight.-Low power consumption.-Compatibility with silicon technology.

It may be possible to produce these devices by means of cheap techniques compatible with industrial scaling-up, such as sputtering or evaporation and condensation methods [78].

MOSs are very popular and widely used in different areas to detect up to 150 different types of oxidizing and reducing gasses and vapours [79].

Research has shown that the gas-sensing process is strongly related to surface reactions. Consequently, one of the key factors affecting gas sensors is the sensitivity of metal oxide-based materials, which can vary depending on several factors that influence surface reactions. These factors include chemical composition, surface modification, microstructures of sensing layers, temperature, and humidity [80].

MOX sensors are based on changes in metal oxide conductivity due to reduction/oxidation reactions taking place on the sensor surface [81,82]. The application of metal oxide gas sensors in Internet of Things (IoT) devices and mobile platforms such as wearables and mobile phones offers new opportunities for sensing applications.

Metal oxide (MOX) sensors are promising candidates for such applications, thanks to their attractive chemical and physical characteristics [83].

Conductometric gas sensors detect the presence of analyte molecules in the atmosphere from variations in their electrical conductance (or resistance). They are based on semiconducting metal oxides, where their electrical properties are modulated via redox interactions with gaseous molecules [63].

Generally, active species, such as O^−^, O_2_^−^, and OH^−^, have been identified as the active centres responsible for the above redox reactions [84]. Such species cover the oxide surface with their relative population depending on the oxide temperature and atmospheric composition [85,86].

In the typical temperature range of metal oxide chemiresistors (200 to 500 °C), O_2_^−^ ions are the most abundant at low temperatures (below 300 to 350 °C), while a higher temperature favours the dissociation of molecular oxygen, leading to atomic oxygen ions O^−^ [82]. The working principle of this technology is that the variation in gas concentration ΔCgas leads to a change of electrical conductivity Δσ, and thus a change in resistance ΔR, as shown in Figure 6. In other words, the principle on which the MOS is based consists of a variation of the conductivity of the metal oxide in the presence of volatile compounds compared to the value assumed by the same conductivity in reference conditions.

#### AI Applied to MOX Sensors Technology

Multi-sensing systems based on gas sensors can be used in several fields of application from quality assurance in the food chain to discrimination between non-chemical fertilized vegetables and fertilized ones [87]. Given the difficult classification task, artificial intelligence (AI) plays a fundamental role due to its ability to process and interpret complex data patterns, enabling more accurate and efficient analysis of scent or odour information [87].

Most of the data pipeline in this domain can be summarized in the following image (Figure 7):

The pipeline is simple: the data are acquired from the MOX sensor, then filtered and the features are extracted, then the dimensionality of the dataset is reduced, then an AI algorithm is implemented.

The AI models in the scientific literature are simple and effective, given the power of AI models to recognize different patterns in the large amount of data created by the MOX sensors. Given the advancement of the cloud and IoT technology in recent years, a system based on MOX sensors coupled with an AI model stored in the cloud could be used as an alert system to ensure food chain quality [88].

## 3. Food Nanotechnology and Its Application Sectors

Nanotechnology applications are currently being researched, tested, and in some cases already applied across the entire spectrum of food technology, from agriculture to food processing, packaging, and food supplements [89].

E-sensing, in particular the e-nose, is used to:Ensure the safety of the product.Follow microorganism growth: sensors are able to track the development of target microorganisms or microbial population over time (hours, days, etc.) based on the release of specific volatile molecules. Generally, VOCs are associated with a determinate concentration of microorganisms (CFU/g or CFU/mL) and by using VOCs’ fingerprints, sensors are able to predict the evolution of microbial ecosystems in various matrices or environments.Recognize the freshness of food products.Detect chemical and microbiological contamination.Detect possible fraud (control of origin and adulteration).Check the authenticity of a product: identify specific geographical areas in support of the tradition of foodstuffs.Verify the level of aroma intensity.Control water and environment.Verify in real time the preparation or storage of food in a domestic or industrial environment [90].

To better understand how nanotechnology works in the food chain, some of its applications for certain foods (oil, coffee, beer, cereals) will be shown below and how they acts in monitoring the development of microbiological and fungal contamination in agricultural fields.

### 3.1. E-Sensing and Food

#### 3.1.1. E-Sensing and Extra Virgin Oil

Olive oil has unique organoleptic attributes, and its consumption is associated with nutritional and health benefits, which are mainly related to its rich composition in phenolic and volatile compounds. The use of olive oil in heat-induced cooking leads to a deep reduction of phenolic and volatile concentrations and changes in the sensory profiles [91].

According to legislation [91], the extension of the thermal degradation of vegetable oils is usually assessed based on the contents of total polar compounds or triacylglycerol oligomer fractions, whose formation is low in olive oils in comparison to other vegetable oils [92]. Although most of the official methods for assessing the quality of oils and fats are relatively simple, some of them are time-consuming and quite expensive, often requiring the use of toxic chemicals and solvents [93,94]. Thus, sensor-based techniques coupled with chemometric tools, e-noses, and e-tongues have emerged as alternative, cost-effective, portable, reliable, and robust approaches to olive oil chemical and sensory analysis [95].

The use of these electrochemical tools has been successfully demonstrated for assessing olive oils’ geographical, cultivar, chemical, and sensory (positive and negative attributes) qualities; detecting olive oil adulterations with other vegetable oils or with low-quality olive oils; quantifying total polyphenolic, flavonoids, and phenolic acid contents in olive oils; and monitoring olive oil quality physicochemical changes during storage [96,97].

Many studies have been conducted to evaluate the characteristics of olive oil, being a product extremely sensitive to light, temperature, and containers.

The quality of oil stored in different containers (opaque crystal bottle; opaque green polyethylene bottle; tin containers) over different conservation times was evaluated in [98] (t0–t6). GC-MS, e-noses, and panellists were used for analysis.

Gas chromatography was used to detect the chemical compounds responsible for the rancid taste of oil samples stored in transparent or translucent containers. Those most responsible for the off flavours in olive oils are aldehydes, ketones, and alcohols [99].

As for aldehydes, (E)-2-esenal, associated with fruity attributes [100], is the most common, and its concentration decreased during storage in all containers. Hexanal and nonanal were identified as compounds associated with increased defects during storage. At t0, the presence of these compounds was not detected in any of the containers, but the hexanal concentration increased in dark glass bottles from t4 and in green PET from t5. These aldehydes are produced via the oxidation of unsaturated fatty acids; this leads to a loss of quality and causes a rancid taste and smell. The second family most represented in fresh olive oil is that of alcohols. 2-penten-1-ol, (E)-2-esen-1-ol and (E)-3-esen-1-ol are related to fruit and the green smell of fresh oils. Generally, the concentration of the first two does not vary over time, unlike the decreasing (E)-3-esen-1-ol. Although hydrocarbons have a negligible impact on the smell of olive oils, compounds such as 4-methyl-tetrane and 2,4-dimethylene, which are related to negative attributes, are increased during storage. Instead, regarding ethers and esters, they do not change concentration during storage [99].

Virgin olive oil was studied also via e-nose. The electronic nose data consisted of an adsorption phase in the static headspace, where the aroma of the samples was brought into contact with the electronic nose sensors for 60 s, followed by the desorption phase to return the gas sensor signal to the baseline. The electronic nose recorded the data at intervals of one second, and the system read the resistive value provided by each sensor. The sensor readings were read by a microprocessor. The resulting data was then sent to a computer and arranged in columns in a spreadsheet for chemometric analysis.

In fact, e-nose data of each container type were processed using principal component analysis (PCA) with the objective of exploring whether there were any natural groupings over the storage time. The PCA results showed that more than 70.0% of the total variance of data was explained by only PC1 in all the containers studied. The obtained results align with the sensory analysis conducted by the tasting panel except for a result in which the electronic nose rightly identified oils after 6 days of storage virgin olive oil (VOO) compared to the panel test that deemed it EVOO. The main difference between these two types of oil is given by free acidity in the form of oleic acid. For EVOO, the free acidity in the form of oleic acid does not exceed 0.8%, while for VOO free acidity does not exceed 2.0% (Figure 8). The increase in free acidity determines a series of modifications, causing a deterioration of the organoleptic characteristics of the oil [101].

Therefore, the e-nose effectively sorted the aroma profiles of the VOO into distinct groups, which highlights its analytical capabilities. The results provided by the tasting panel and the volatile compound profile matched the classification provided by this device. This proves that, combined with chemometric tools, the e-nose represents a fast, simple, reliable, and low-cost method suitable for use at the industrial level to control the quality of this product. Therefore, olive oil is a product extremely susceptible to organoleptic changes due to many factors, and according to its free acidity, it can be classified into various categories of value [102]. Extra virgin olive oil is the best vegetable oil but, at the same time, is one of the products that is commonly a victim of fraud in the agri-food sector. Several scientific techniques have been applied in order to guarantee the authenticity and quality of EVOO [102]. Most of the volatile compounds are synthesized via the interaction between enzymes and substrates during the olive fruit crushing, and possible synergistic or antagonistic effects also come into play [103].

In addition to being used to define the quality of the oil, the electronic nose can be exploited along with other technologies to verify if the oil has been adulterated.

The other specific technologies to evaluate the quality of edible oils are:Thermometry [104]Machine vision [105]Spectroscopy [106]Ultrasound [107]Sensory method by trained people [108]

Blending oils of mean and low nutritional value with extra virgin olive oil causes changes in chemical and physical properties. Electronic noses or electronic tongues could be used for the study of the chemical components; moreover, an ultrasonic diagnostic system with a machine learning algorithm can detect differences in physical properties, such as density and homogeneity, which has a direct effect on the velocity, attenuation, and refractive index of waves [109]. Studies have been conducted to classify extra virgin olive oil using two powerful methods, ultrasound and an e-nose device with eight semiconductor metal oxide gas sensors. Combining the data from both systems, the accuracy of classification increases in addition to the results being validated. Also, by using the two techniques, the weaknesses of one method is compensated for by the other method [110]. The ultrasound system provides several peaks caused by the impact and reflections of transmitted ultrasonic waves that experience different environments. The raw data were transferred to a computer and converted to data that could be processed using MATLAB R2014a software. Instead, the e-nose device was employed to capture the olfactory characteristics of both extra virgin olive oil and fraudulent samples. These characteristics are distinctive and characteristic for each sample, rendering them unique identifiers [111].

The results showed that the data obtained through the ultrasonic system outperformed the e-nose data. The ultrasonic method measures the internal characteristics of a material including its density, texture, and chemical composition, whereas the e-nose system judges a material based on the volatile compounds of the sample. In this research, the ultrasonic characteristics held greater efficacy. Consequently, the structural and textural properties of the substance play a more crucial role than the gasses emitted from it in detecting a liquid substance such as olive oil. Nonetheless, it is worth noting that employing a combination of both methods enhances confidence and reinforces the validity of the results [112].

#### 3.1.2. E-Sensing and Coffee

Coffee is one of the most consumed beverages in the world. Moreover, it is the second most consumed beverage after water, and its consumption is constantly increasing [113]. Coffee brewing methods can be changed depending on the geographic, cultural, and social environment as well as individual preferences. There are different methods to extract and brew coffee (boiled coffee, filtered coffee, moka coffee, espresso coffee, etc.). Therefore, coffee processing is an important factor and should be studied in detail. The aroma is one of the most important coffee attributes, and it depends on the species, variety, and fruit, climatic and soil conditions, cultivation, and post-harvest storage [114]. Reactions of decomposition of non-volatile compounds contained in raw coffee, pyrolysis, caramelization, and Maillard reactions yield the final aroma [115]. Due to the complexity of the coffee aroma, a variety of applications of the electronic nose have been carried out in past years [116].

Some studies have been conducted to identify the composition of the volatile profile. There are many methods for the assessment of aroma: organoleptic and instrumental methods, gas chromatography (GC-MS), and chemical determination of the composition [117,118,119]. In addition to these methods for evaluation of the aroma and chemical composition of biomaterials, electronic nose devices have been increasingly used in recent decades as fast, simple, and non-invasive tools for the assessment of the quality of biological products [120].

A study was conducted using an electronic nose to discriminate between the automatically volatile compound profiles of coffee from different countries, roasted under identical time and thermal regimes. This inexpensive device was used for rapid detection of aromas. The results of the e-nose analyses were verified using the more accurate but more laborious and expensive GC-MS technique for the analysis of volatile substances [116].

The investigations demonstrated that the degree of roasting and the type of device used did not alter the individual aromatic properties of the analysed coffee. The volatile compound profiles and the content of their main groups were specific to each coffee type and were associated with the conditions of growth, harvesting and storage.

The analysis of volatile compounds carried out with the use of the Agrinose yielded results consistent with those obtained with GC-MS. Quick and low-cost analysis of volatile substances in coffee using an electronic nose with a matrix of MOS sensors is a reliable tool for assessment and classification of coffee types [116].

Other studies were conducted with the electronic tongue to verify that the product had not been subjected to adulteration. In fact, due to its high commercial value, ground roasted coffee has been the subject of frequent adulterations. In summary, coffee adulteration can be carried out in two ways: by changing the quality of beans (different species, geographical origin, and defective beans) and by adding other materials of low economic value, high availability, and similarity to coffee after roasting and grinding [121]. Major coffee adulterants include coffee processing by-products, such as coffee husks and sticks, as well as spent coffee grounds, brown sugar, and grains [122]. The conventional method used to identify and quantify adulterations in ground roasted coffees is based on optical or electron microscopic analysis of chloroform pretreated coffee powders. Such analyses are slow and subjective and could generate contradictory results [123]. Much effort has been devoted to the development of more reliable and reproducible methods to identify coffee adulteration. Electrochemical methods have shown to be rapid, low-cost, and efficient alternatives for the evaluation of coffee quality, mainly comprising an array of electrochemical sensors aided by multivariate analysis, also known as the electronic tongue [124]. Examples include discrimination between Arabica and Robusta coffees [124], civet coffee [125], Robusta coffee cultivars with different roasting degrees [126], and identification of coffee features [127]. Thus, the aforementioned findings show that the e-nose and e-tongue are very promising devices for the monitoring of the quality of coffee.

#### 3.1.3. E-Sensing and Beer

Beer is the most produced and consumed alcoholic product in the world, with 177.5 million kilolitres being produced every year [128].

The taste and smell of different beers can vary significantly due to distinct manufacturing processes. Because of these variations in manufacturing, beers come in a wide range of flavour profiles and aromas [129]. Some beers may have a bitter taste, while others may have a sweeter or malty flavour. Similarly, the aromas can range from floral and citrusy to roasty and chocolatey [130]. This is why each beer can be seen as having its unique character, and this individuality is one of the appealing aspects of beer for enthusiasts and connoisseurs [131]. This diversity is particularly prominent in the craft beer industry, where small, independent breweries often experiment with various ingredients and techniques to create distinct and innovative beer offerings [132,133]. Nanotechnology and nanosensors could be applied to analyse beer. With the development of the electronic nose technique, it has been possible to obtain analytical responses in a few minutes and from the overall volatile composition of the samples [134]. However, the flavour features of beer are complicated due to its composition and concentration. Therefore, combining e-tongue and e-nose technologies in a fusion system can efficiently and comprehensively capture the taste and olfactory characteristics of beer, which is valuable for various applications in the beer industry, including quality control and product development [135]. In fact, it has been shown that compared with the single e-tongue and single e-nose, the classification accuracy rate of beer flavour information was improved by using multi-sensor data fusion.

The SA-402B e-tongue was employed to gather beer taste information. This device comprises a sensor array, an automated detection system, a data acquisition setup, and specialized data analysis software. The sensor array is composed of five taste sensors, and each sensor is composed of a unique artificial lipid-based membrane [129].

The electronic nose used was the PEN3 e-nose. The instrument includes a gas collection device, a gas detection unit, and an air purification device. The gas detection unit includes a sensor array and a pattern recognition analysis and processing system. The sensor array contains 10 metal oxide gas sensors, which can achieve the detection of olfactory cross-sensitive information. The specific components targeted for detection by these sensors are as follows: aromatic (W1C), hydrocarbon (W5S), aromatic (W3C), hydrogen (W6S), broad-methane (W1S), sulphur-organic (W1W), broad-alcohol (W2S), sulphur-chlorine (W2W) [129].

Another study was conducted using e-nose and mass spectrometry instead of electronic nose and electronic tongue: MS-e-nose could be a potential aroma sensor because it can discriminate and characterizing the samples according to their predominant aromas with the help of the techniques of multivariate analysis.

Moreover, with the application of variable selection techniques, it is possible to obtain information about the compounds possibly responsible for differences between the samples [132]. In addition, e-nose was used to establish the acceptability of foods by consumers. In fact, through beer’s alcoholic fermentation, many different volatile compounds are produced, which starts from the interaction between yeast and the matrix. Molecules such as diacetyl or 2,3-Butanedione have a great impact on the flavour of many foods, from dairy products to fermented matrices including beer, thanks to their small and highly volatile characteristics. Because of its butter-like aroma, it is commonly used as a flavouring agent, and it is considered as GRAS by the FDA [133,134,136].

Like many other molecules, it can be an appreciated characteristic of some food products or become a problem if its presence is detected in others [134].

Implementing a detection system that is able to recognize diacetyl’s presence is of primary importance. Chromatographic analyses are the most used techniques to detect diacetyl’s presence in foods because of their good results; meanwhile, other analytical procedures include colorimetric assays and voltametric detection [133], which can guarantee excellent results for miniaturizing samples [134]. Implementing electronic noses for beer evaluation is nowadays a well-known field, as this topic has been investigated in the last 30 years [136,137]. Nowadays, many studies are currently underway regarding the creation and validation of IoT-integrated devices that are able to detect different target molecules in fermented products [138].

The table below shows some examples of sensors and sensitive elements used in the food industry for the recognition of specific molecules (Table 1).

#### 3.1.4. E-Sensing and Cereals

Cereals and cereal by-products constitute a major part of the daily diet of human and animal populations. Among the most important risks associated with cereal consumption are mycotoxins. Mycotoxins are fungal secondary metabolites that have a great impact on human and animal health. Moulds that can produce mycotoxins grow on numerous foodstuffs such as cereals, dried fruits, nuts, and spices. They can cause a variety of adverse health effects and pose a serious health threat to both humans and livestock. The adverse health effects of mycotoxins range from acute poisoning to long-term effects such as immune deficiency and cancer [139]. It has been estimated that up to 25% of the world’s crops grown for food and feeds may be contaminated with mycotoxins [140]. More than 300 mycotoxins are known; however, aflatoxins, trichothecenes, zearalenone, fumonisins, ochratoxin A, T-2 and HT-2 toxins are the main contaminating mycotoxins in food [141,142,143]. Developing reliable and rapid methods for mycotoxin detection is a priority (Table 2) [141]. Legal limits for the maximum content of mycotoxins allowed in different matrices have been set in most countries worldwide [144]. The e-nose has been identified in many studies as an effective tool for rapidly screening food substances [145].

The main application areas of e-nose analysis to investigate the causes of cereal damage are reported below:Detection of mycotoxigenic in contaminated grains;Detection of insect odours in grains;Semiquantitative/quantitative evaluation of mycotoxins in contaminated grains;Detection of VOCs as indicators of potential grain spoilage.

Detection, identification, and quantification of plant diseases via sensor techniques are expected to enable more precise disease control, as sensors are sensitive, objective, and highly available for disease assessment [146].

Electrochemical (EC) biosensors based on graphene nanomaterials for mycotoxin identification have recently been developed [147]. They do not need reagents for analysis, require minimum sample preparation, and can directly detect biological matrices. Furthermore, they have an affinity for mycotoxin detection [148], a fact that indicates their importance in this area. EC sensors operate on the fundamental principle of measuring the potential difference between the transducer and the electrode interface [149]. One of the main disadvantages of using EC sensors is low specificity and selectivity. However, the integration of molecular recognition elements into EC sensors enhances their efficiency by enabling the accurate detection of target molecules. The primary categories of recognition elements suitable for achieving high sensitivity in EC biosensors include molecularly imprinted polymers (MIPs) [150], aptamers (APTs) [151], and antibodies [152]. These works have highlighted the potential of these sensor types in detecting a wide array of molecules and foodborne toxins, including mycotoxins. In comparison to antibodies, aptamers exhibit notable advantages. They possess physically persistent and extreme pH or temperature treatments that do not interfere with nucleic acid refolding ability, whereas most antibodies are susceptible to complete degradation under such conditions. Their cost-effectiveness, simplicity of operation, and ease of integration make EC-based aptamers highly suitable for mycotoxin detection [153]. It is worth noting that the performance of these aptamer-based sensors depends on both the configuration of the recognition element and the choice of the transduction technique [154]. The combination of various graphene nanomaterials improved EC sensors’ efficiency, enabling them to detect-mycotoxins with high selectivity and sensitivity [155].

As regards MIPs, they are polymeric matrices with cavities with an affinity for specific molecules. MIP formation is based on polymerization, which involves one or more functional monomers in conjunction with a cross-linker. After polymerization, the target molecule is extracted from the polymer matrix, resulting in the creation of cavities within the structure. These cavities possess sizes, shapes, and interaction properties that are complementary to the template molecule. The MIP-based sensors’ efficiency is related to the number of imprinted cavities present on the sensor surface. While numerous electrochemical (EC) sensing tools for mycotoxins have been developed, they have predominantly remained confined to laboratory-scale applications, rendering them inaccessible to the majority of end-users. Consequently, future research in EC sensors should prioritize the development of practical, portable EC devices capable of multi-mycotoxin detection. Such advancements hold promise for applications in the realms of food quality control, food processing, and manufacturing [156].

**Table 2 sensors-23-08429-t002:** Some techniques to identify mycotoxins in food.

Mycotoxin	Technique	Working Range or Sensitivity	Limit of Detection	Food Product	Reference
AFB1	IC/IP-AMP on 96-well SPEs	0.05–2 ng mL^−1^	0.03 ng mL^−1^	Corn	[148]
HT-2	IC/IP-AMP on 96-well SPEs array	1.2 ng mL^−1^	0.2–	Corn	[151]
AFM1	DNA/AMP	1.9–20.9 nM	/	/	[147]

Abbreviations: AFB1: aflatoxin B1; IC: indirect competitive assay; AMP: amperometry; AFM1: aflatoxin M1.

## 4. Robotics Applications and Smart Irrigation in Agriculture

In the agri-food sector, nanotechnology is heavily used, but robotics is also gaining ground lately. The objective of agricultural robotics is to help the sector in its efficiency and the profitability of the processes. In other words, mobile robotics works in the agricultural sector to improve productivity, specialization, and environmental sustainability. Labour shortages, increased consumer demand, and high production costs are some of the factors that have accelerated automation in this sector, with the aim of reducing costs and optimizing harvests [157,158].

Robots can be useful tools to solve many problems and simplify particularly heavy jobs, such as:-Chemical application, spraying, or harvesting, as required by the fruit or plant;-Collection and conversion of useful information for the farmer;-Selective application of pesticides;-Selection to avoid food waste;-Optimize the harvest and production of raw materials.

Robotic systems have made it possible to mitigate the risk of surface bruises and ruptures, destruction via crushing of plant tissue, and plastic deformation in the harvesting of fruits with a soft rind, such as apples, cherries, pears, stone fruits, kiwifruit, mandarins, cucumbers, peaches, and pomegranates.

Smart irrigation systems powered by renewable energy sources have been proven to substantially improve crop yield and the profitability of agriculture [159,160,161]. The control and monitoring of a solar-powered smart irrigation system can be achieved using sensors and environmental data. The predicted values of water level, weather forecast, humidity, temperature, and irrigation data are used to control the irrigation system. Even though smart farming technologies, which were developed to mimic nature, could help prevent climate change and enhance the intensification of agriculture, there are concerns about long-term ecological impact, cost, and their inability to complement natural processes such as pollination [162]. Despite these problems, the market for bio-inspired technologies with potential agricultural applications to modernize farming and solve the abovementioned challenges have increased exponentially. Smart farming technologies have been proven to reduce production costs and improve yields via the intelligent regulation of humidity, irrigation, frost, greenhouse microclimate, and pesticide and fertilizer applications [163].

Robots can also be used for other purposes than those mentioned above. Nowadays, robots are often implemented for volatile organic compounds (VOC) detection, implementing these devices for leak detection and solving gas-related problems. MOX sensors can be implemented on devices able to freely move in an environment. This has already been implemented in the oil and gas industries, but a large development field is represented by primary production, where robots are used for crop condition identification, to collect information from the environment of an olive grove to maximize the olive yield, or to compare different types of wine from vines subjected to different climatic conditions [164].

MOX sensors, although excellent for sensitivity, reliability, low deployment cost and low complexity in electronic parts, require a second electronic circuit to heat up beside the sensing circuit. Moreover, drones require important energy consumption to operate. Thus, in order to properly take advantage of this technology, power usage is crucial.

A study was conducted to detect contamination by pathogens in tomatoes. The analyses were carried out using GC-MS and an e-nose with SMO sensors, produced via the process of sputtering a thin layer of sensing material onto an alumina substrate. E-nose outputs olfactory patterns in a multidimensional feature space; therefore, multivariate analysis and pattern recognition techniques must be used for visualizing and analysing the data. It is necessary to use principal component analysis (PCA) for an unsupervised exploratory data analysis and k-nearest neighbours (kNN) for classification purposes.

## 5. Smart Farming

### 5.1. E-Sensors for Smart Farming

Nowadays, farming is an important field of development because, through innovation within this field, it is possible to pursue important goals in terms of both reduction of carbon footprint and overall final quality improvement. Increasing the efficiency of production is an important aspect of facing continuous growth of meat consumption and the increasing population [165]. Several studies have demonstrated how meat consumption and technologies in animal rearing are both necessary in order to reach all the different fixed targets, such as the aim of the European Union to be climate neutral by the year 2050 [166].

### 5.2. Sensors for Ammonia Detection

Livestock production, especially when concentrated in small areas, has a direct and important impact on ammonia and greenhouse gas emissions and its reduction by 50% from the 2018 levels by the year 2030 is one of the goals imposed by several countries, like Netherlands [167]. Ammonia is a toxic compound typically produced in biological and industrial processes. Around 80% of NH3 is produced for nitrogen-based fertilizer. Maximum safe levels are set to 25 ppm for long exposure, namely more than 8 h, or 35 ppm for shorter exposure, defined as 15 min maximum [168]. Because of the importance of this topic, many researchers have implemented several detection methods to manage its production, alongside several ways to reduce the production, such as introducing better management on recoupling livestock and feed production. A 47% reduction in greenhouse gas emissions and 27% of ammonia emissions was shown in [167]. Gas sensors are often implemented for hazardous gas recognition, as they show excellent recognition properties for gasses such as CO, CO_2_, NOx, SOx, and NH_3_. There are 3 main NH_3_ sensing techniques: solid-state sensing methods (e.g., MOX sensors); optical methods (e.g., diode laser spectroscopy), and other methods (e.g., electrochemical sensors) [169]. MOX sensors are implemented because of their simplicity, low cost, and flexibility. SnO_2_, ZnO, TiO_2_, WO_3,_ and MoO_3_ are the most implemented metal oxides for NH_3_ detection [170]. On the other hand, metal oxides show low selectivity in detecting specific gasses from a mixture [171], which can generate problems for in-field recognition. Nevertheless, they are nowadays the most implemented sensors for NH_3_ recognition. For in-field applications, MOX sensors can be improved in order to reduce their operation temperatures. This is mainly possible due to increasing electron mobility via the implementation of carbon nanotubes or layered graphene [172].

### 5.3. Sensors for Animal Welfare

Implementation of technology has also the aim of improving welfare. Animal welfare has a direct impact on the organoleptic properties, and it goes through all the actions and choices made during the breeding [173]. This aspect has to be considered as a requirement to obtain quality certification, which is now essential to interfacing in a complex, globalized, and competitive market. Animal welfare is mainly distinguished by good feeding, good housing, good health, and appropriate behaviour [174]. Decent results on these main classes can be reached via the compliance with different criteria, which can be managed with the implementation of an IoT system able to collect and administer heterogeneous data from different sensors [175]. Sensors can be implemented in an invasive way, directly monitoring when swallowed or implanted on specific animals, or a non-invasive way which includes cameras and other sensors to control environmental parameters in the farm [176]. These software-based solutions can self-regulate behaviour by monitoring different parameters related to the environment, to the animals’ health condition, and to predicting unfavourable conditions [177]. The main controlled parameters are described in the following table (Table 3):

Sensors in farm industries have been implemented for almost a decade [165], but different technologies have already been implemented in different ways in animal rearing and management before [166,167]. Technology has changed the everyday work in the farmyard, reducing manual and repetitive tasks [176] and improving speed, cleanliness, and overall quality. As described in Table 4, many different technologies can be applied for precision livestock welfare. Accelerometers are the most implemented sensors, followed by cameras, load cells, and miscellaneous milk sensors [187].

The correct implementation of veterinary medicinal products has an important impact on animal welfare and good farm management because treating diseases is often expensive and time-consuming and can generate situations where part of the herd of the animals have to be slaughtered. In contrast to these reactive procedures, the implementation of sensors in the farm industry can predict illness before the most serious symptoms appear [197]. This can produce a reduction of cost and the possibility of improving the capability of facing serious hazards by implementing quarantine for specific animals. For small animal production, the implementation of an IoT system can induce important changes, as normally these animals have been treated as lots instead of single units [168]. This induces a change in the way farms operate, reducing costs and avoiding the possibility of rejecting animals, as curing only sick individuals is uneconomical.

#### 5.3.1. Sensors for Disease Control

Technology applied alongside good manual practices in the farm industry plays an important role in reducing risks of animal death, crop failure, and, as a consequence, financial loss [177]. Disease management represents one of the most useful and important research fields where this technology is studied and implemented, in particular for production where promiscuity among animals is high, e.g., poultry farming [198]. More generally, all animal production has to constantly face diseases, but nowadays thanks to IoT processes and sensor implementation inside farms it is possible to quickly detect illnesses such as mastitis, lameness, postpartum disease, coccidiosis, and African swine flu [186].

Mastitis prevention is an important development field for IoT system implementation, as this disease caused by bacteria deeply affects the economy of the farm [198,199]. The implementation of a system able to recognize its presence with a high degree of accuracy, preventing it and constantly monitoring is an important field of development where different technologies are implemented [200]. According to the seriousness of the mastitis, different approaches based on several sensors must be implemented, which vary depending on sensitivity and specificity in different time windows [201]. The first commercially available sensors for mastitis recognition were mainly based on electrical conductivity [202,203], while nowadays enzymatic reactions, somatic cell count or colorimetric assays and homogeneity tests are the most implemented solutions. It was reported that electronic noses and volatilomics can be used to investigate mastitis presence [204,205,206], allowing for early detection via cost-effective solutions. Mastitis has also an important role in animal social behaviours, and monitoring position in the herd can give important information when analysed via artificial intelligence algorithms [190]. Some research studies have been carried out for 4 different mastitis situations: cows with severe clinical mastitis needing immediate attention; cows with subclinical mastitis not needing immediate attention; cows needing attention at drying off; and monitoring of udder health at the herd level. Each of these 4 mastitis management area has specific demands for a sensor system. Consequently, to achieve a comprehensive and effective deployment of sensor-driven mastitis management, rather than relying on a single generalized alert algorithm, as is the current practice, the algorithms of these sensor systems should be tailored to meticulously monitor each of these four distinct mastitis scenarios [207].

Lameness poses a substantial issue for both performance horses and farmed animals, resulting in considerable implications for animal welfare and treatment expenses. Currently, lameness diagnosis primarily relies on subjective scoring techniques conducted by trained veterinary clinicians. However, the implementation of automatic methods utilizing appropriate sensors would enhance both efficiency and reliability. Its recognition is based on kinetic and kinematic measurements and they are more often implemented with radar sensing [208], accelerometers, cameras [209], and electromyography [210]. Moreover, indirect analysis such as behaviour control can be successfully implemented. Indeed, by analysing the time spent standing, lying, or walking, it is possible to quickly recognize this issue and maintain high productivity on farms [211].

A considerable number of lactating dairy cows experience diverse disorders during the early postpartum period, leading to detrimental impacts on their health, welfare, reproduction, and performance and resulting in substantial economic losses [212]. Analysing real-time parameters such as localization, heat detection, activity measurement, rumination monitoring, and parturition in dairy cows can give important information on many diseases with important repercussions on animal wellness and productivity [213].

Coccidiosis remains one of the prominent parasitic infections in the poultry industry. This infection is attributed to protozoa of the Eimeria genus, resulting in significant economic losses due to malabsorption, impaired feed conversion rate, decreased weight gain, and heightened mortality rates. The most severe consequences are observed in commercial poultry farms where birds are raised in close proximity and high densities [214]. Because its diagnosis is difficult, preventive actions such as chemoprophylaxis or vaccines are extensively implemented. Novel, highly sensitive, and rapid diagnostic methods have been investigated to assess the feasibility of diagnosing diseases in both livestock and humans. These innovative techniques rely on the detection and analysis of VOCs generated by pathogens, interactions between hosts and pathogens, and biochemical pathways [215]. The study of VOCs in the environment is therefore of fundamental importance despite being imperceptible by humans. Environmental air monitoring can be performed by instruments such as the electronic nose, which captures volatile molecules even when the threshold of human perception does not reach.

African swine fever is a viral haemorrhagic disease with extremely important repercussions on the farm economy, as its mortality is extraordinarily high [216]. Early detection of African swine fever can be achieved by capturing video footage using a camera and processing it with an optical flow algorithm. The underlying assumption of this research is that infected swine would exhibit reduced movement speed [193].

#### 5.3.2. In-Field Control Challenges

Currently, the most important barrier to the spread of advanced technology such as sensors, blockchain technology, or big data analysis mainly consists of the actual integration of the several components needed. Indeed, environmental conditions, and communication devices have to be complied or implemented. Moreover, sensors have to be online and have to work even at long distances [217] and developers can have limited experiences in farm industries, where real applications are often far from the ideal conditions found in the laboratory. This is even more important if the implementation of IoT-based technologies is made in developing countries, where yield per hectare is much lower and the largest investments have to be accomplished [218].

Understanding animal behaviour through data imaging is nowadays a field where important steps have to be taken, as different emotions are misinterpreted [219]. Nevertheless, the integration of this advanced technology into traditional processes can bring important benefits for resource management, less area to be designated for animal growth and animal welfare for the same production quantities [220]. This allows the introduction of precision farming, where animals and the environment can be divided into subclasses in order to have better management [221]. This can generate sustainable and long-term positive value bringing reproducibility, authenticity and tracking of data generated by several sensors [222]. As an example, early detection of disease helps to separate only specific animals, as shown in Figure 9.

### 5.4. IoT for Green Energy Production-Biogas Production and Management

Green energy is a term that identifies a vast and heterogeneous group of different energy resources, mainly characterized by a reduced carbon emission, unlike conventional energy based on fossil fuels [223], and nowadays it has become predominant. Biogas production has been exploited for many years, as it provides costless energy starting from waste fraction [224,225]. Nowadays, biogas production can be highly topical because its development and implementation can help to reduce reliance on countries which are natural gas suppliers. The possibility of having delocated, easy-to-build and operate, costless power plants could be a reasons for implementation in the farm industry [226]. Moreover, thanks to anaerobic co-digestion, it is also possible to manage farm industry by-products, controlling odours, and reducing greenhouse gas emissions [227]. In the anaerobic digester, H_2_, CH_4,_ and CO_2_ are constantly monitored to provide information about the process productivity. Moreover, pH and volatile fatty acid concentrations are other important indicators for correct methane production [228].

Several sensors can be implemented to monitor the production efficiency, standing out for the necessity to extract samples (at-line) or work directly in the production plant (in-line). In-line sensors should be resistant to pressure, temperature between 35 and 55 °C, cleaning processes, and the response time [229]. Several monitoring procedures are based on classical laboratory analysis, e.g., titrimetry [230], gas chromatography, and near-infrared and infrared spectroscopy [231]. On the other hand, monitoring can be based on algorithm-based systems [228], implementing MOX sensors or colorimetric sensor assays. Introducing IoT systems help modernize the energy industry both on safety and efficiency criteria [232], reducing process deterioration and increasing the efficiency of production. MOX sensors are able to recognize the presence of gasses [233] and e-noses have been already implemented to monitor biogas production, supporting circular economy [234,235]. Gas recognition through MOX sensors has also been exploited for other purposes, e.g., the safety of workers and workplaces [236], which demonstrates the capability of this technology to monitor methane emissions at different concentrations. VOCs have been recognized at low concentrations, from 2 to 10 ppm, demonstrating the application possibility of these sensors [237,238]. Meanwhile, high concentrations can also be a challenge, as sensing elements can be saturated by these substances. Studies reported specifically designed p + n field effect transistors to recognize concentrations up to 2000 ppm [239].

Colorimetric sensor assays (CSAs), also known as optical chemosensory nose or tongue systems, are composed of a matrix that retains the indicator near a light source. The obtained interaction changes the optical properties of the indicator in direct proportion to the concentration of the analyte [240]. VOCs can be also detected via CSA, increasing the sensitivity of less-reactive VOCs and improving the limits of detection [241]. Biogas production can be monitored via CSA as an on-line sensing element and outcome data are further analysed with principal component analysis. This technology shows good results, as it can detect strain populations and chemicals present inside the bioreactor such as methanediol, glucose, or fructose, as well as many others [242]. Finally, it is a cost-effective solution easily implemented in power plants [228].

## 6. Conclusions

In summary, consumer acceptance and food safety are key concerns for wholesalers and retailers of foodstuffs. Standard analytical methods to verify products’ freshness assessment are based on the measurement of chemical and physical attributes related to appearance, colour, texture, odour, and taste. These methods have disadvantages, such as being destructive, expensive, and time-consuming. In the last decade, rapid advances in the development of new technologies for evaluating food quality attributes have led to the development of non-invasive and non-destructive instrumental devices, such as biosensors, e-sensors, and spectroscopic methods. Devices similar to human-based senses, such as e-noses, e-tongues, and e-eyes, can be used to analyse different compounds in several food matrices. These sensors allow for the detection of one or more compounds present in food samples and monitor the entire food chain, starting from agriculture and/or farming to the production of foodstuffs. The excellent specificity of the nanosensors allows for an analysis of a wide variety of analytes, including heavy metal ions, toxins, pathogens, small molecules, nucleic acids, and proteins. Innovative devices and modern technologies are being developed that can facilitate the preparation of food samples and their precise and inexpensive analysis. From this point of view, the development of nanosensors to detect microorganisms and contaminants is a particularly promising for their application in food nanotechnology. Surely the future goal will be to make sensors that are the most sustainable. Researchers will try to obtain efficient sensors in the shortest possible time, avoiding the use of chemicals. In fact, nanotechnology holds great promise to provide benefits not just within food products but also around food products.

## Figures and Tables

**Figure 1 sensors-23-08429-f001:**
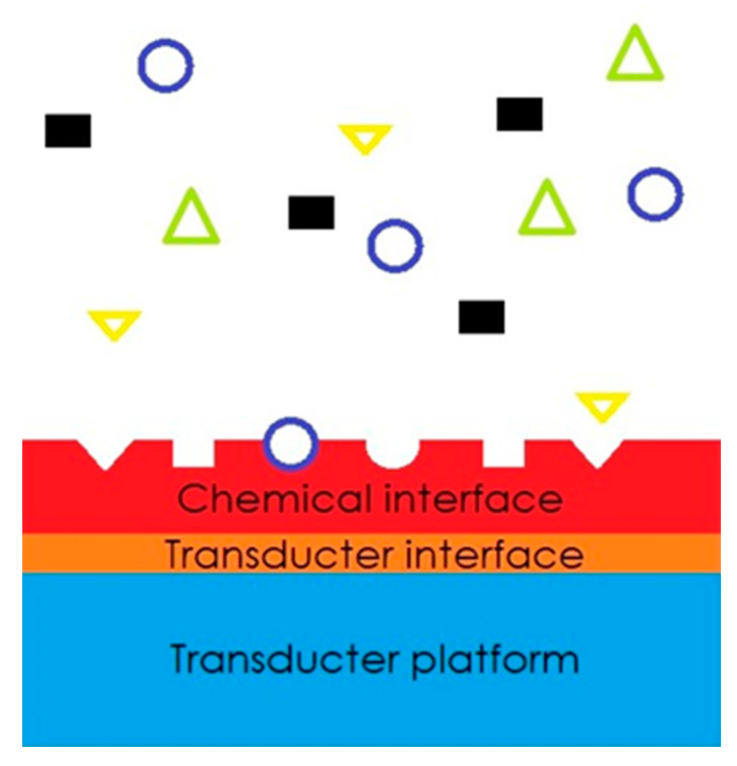
Diagram of the sensor activities.

**Figure 2 sensors-23-08429-f002:**
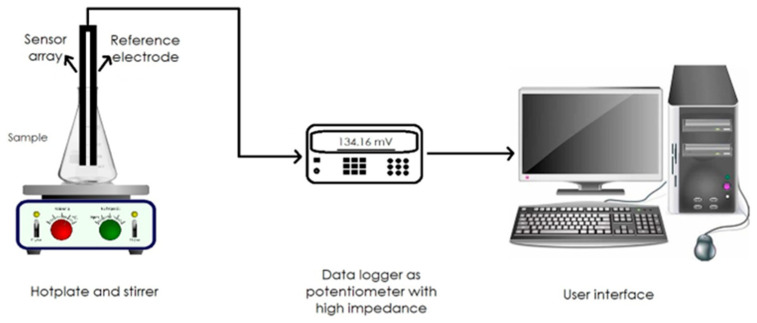
E-tongue setup, including homemade sensor array, data logger, and PC.

**Figure 3 sensors-23-08429-f003:**
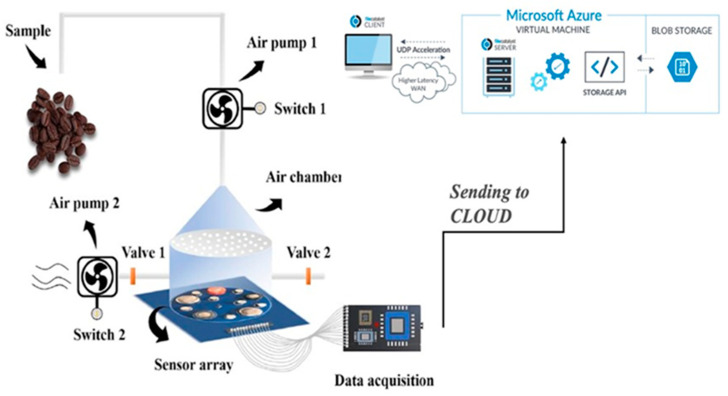
Description of how an electronic nose works.

**Figure 4 sensors-23-08429-f004:**
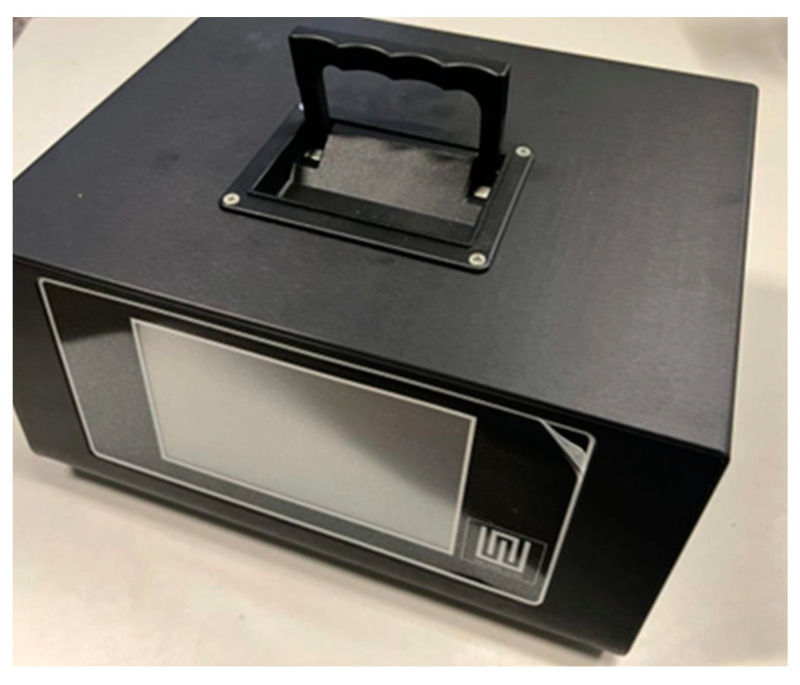
Device S3.

**Figure 5 sensors-23-08429-f005:**
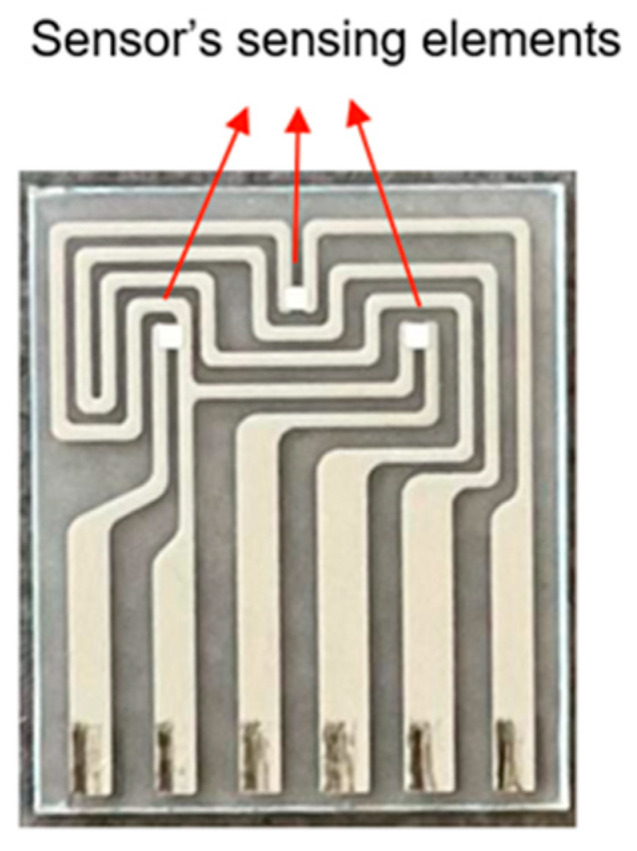
Sensor with three sensitive elements for the volatile compounds’ analysis.

**Figure 6 sensors-23-08429-f006:**
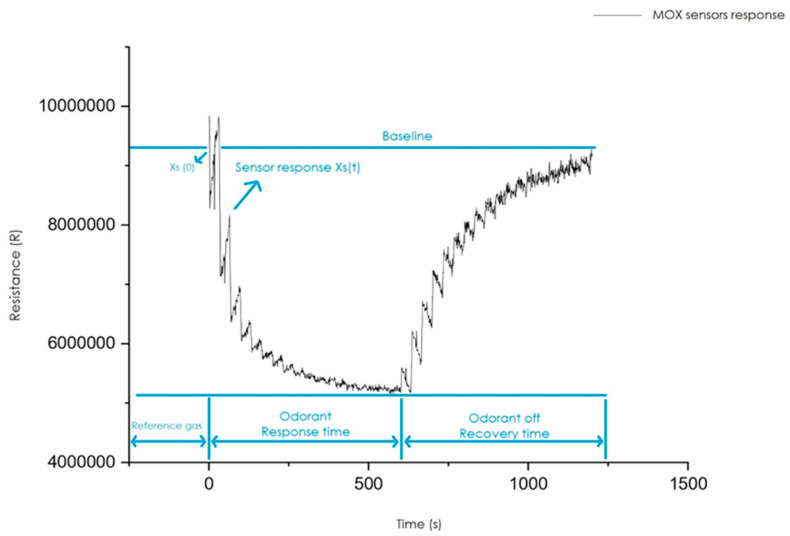
Typical response of a MOX sensor.

**Figure 7 sensors-23-08429-f007:**
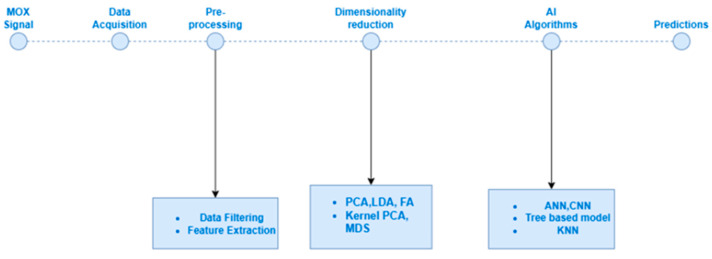
Data pipeline.

**Figure 8 sensors-23-08429-f008:**
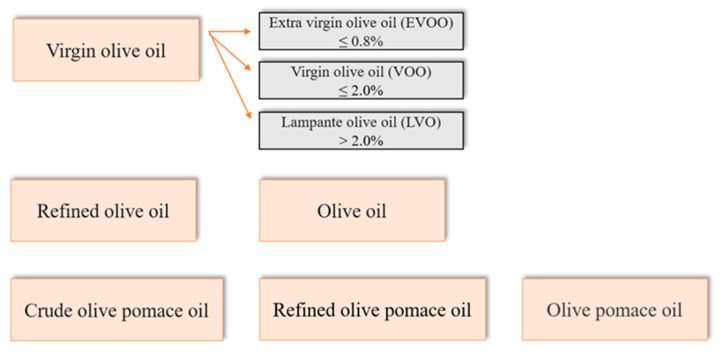
Olive oil’s categories.

**Figure 9 sensors-23-08429-f009:**
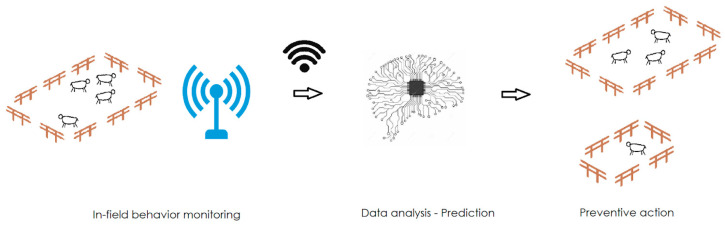
Sensors network able to rapidly detect data from the field which are analysed to produce predictive actions.

**Table 1 sensors-23-08429-t001:** Sensors and sensitive elements used in the food industry for the recognition of specific molecules.

Sensor	Sensing Element	Deposition Technique	Operating Temperature	Gas Analysed	Detection Range	Food Product	Reference
TGS813	SnO_2_	Thick film techniques	200 °C	Methane, propane, and butane	500–10,000 ppm hydrocarbons	Olive oil	[98]
TGS822	SnO_2_	Thick film techniques	200 °C	Carbon monoxide and vapours of organic solvents	50–5000 ppm EtOH	Olive oil	[98]
MQ3	SnO_2_	/	/	Alcohol vapour	/	Olive oil	[100]
MQ4	SnO_2_	/	/	CH_4_	200–10,000 ppm CH_4_, natural gas	Olive oil	[99]
MQ8	SnO_2_	Thick film techniques	400 °C	H_2_, alcohol, LPG, and cooking fumes	100–10,000 ppm H_2_	Olive oil	[99]
MQ136	SnO_2_	/		H_2_S	1~200 ppm H_2_S	Olive oil	[111]
TGS2600	SnO_2_	Thick film techniques	200–400 °C	General air contaminants, hydrogen, and carbon monoxide	1~30 ppm H_2_	Coffee beans	[116]
TGS2602	MOS type	Thick film techniques	200–400 °C	Ammonia, hydrogen sulphide, high sensitivity to VOC, and odorous gasses	1~30 ppm EtOH	Coffee beans	[116]
TGS2603	MOS type	Thick film techniques	/	Odours generated from spoiled foods	1~10 ppm EtOH	Coffee beans	[116]
TGS2610	SnO_2_	Thick film techniques	200–400 °C	LP gas and butane	1~25% LEL	Coffee beans	[120]
TGS2611	MOS type	Thick film techniques	200–400 °C	Natural gas, methane	500~10,000 ppm	Coffee beans	[120]
MLV-P2—CO	MOS type	/	300 °C	Butane, methane, ethanol, and hydrogen. Specifically designed for volatile organic compounds	1~20 ppm	Coffee beans	[116]
W1C	MOS type	Thick film technique	200–400 °C	Aromatic compounds	10 ppb	Beer	[135]
W5S	MOS type	Thick film technique	200–400 °C	Alkenes, aromatic compounds, and less polar compounds	1 ppb	Beer	[129]
W3C	MOS type	Thick film technique	200–400 °C	Aromatic compounds	10 ppb	Beer	[135]
W1S	MOS type	Thick film technique	200–400 °C	Broad-methane	10 ppb	Beer	[129]
W1W	MOS type	Thick film technique	200–400 °C	Sulphur–organic	1 ppb	Beer	[132]
W2S	MOS type	Thick film technique	200–400 °C	Broad-alcohol	10 ppb	Beer	[129]
W2W	MOS type	Thick film technique	200–400 °C	Aromatic compounds, sulphur organic compounds	1 ppb	Beer	[129]

**Table 3 sensors-23-08429-t003:** Parameters controlled and implemented in IoT algorithms for livestock monitoring.

Parameter	Description	References
Absence of prolonged hunger	Image data enables the control of the body condition and the amount of time spent waiting in front of the feeding table when food is unavailable	[178]
Absence of prolonged thirst	Detection made evaluating the period spent at drinkers through GPS and accelerometers, RFID detectors or camera tracking.	[179]
Comfort around resting	Time spent in lying areas can be obtained from real-time location system (RTLS) technologies, the time spent actually lying down can be recorded with accelerometers and heart rate monitoring.	[177,180]
Thermal comfort	Sensors for heart rate monitoring or respiratory rates for large species, environmental conditions monitoring and body temperature can be monitored.	[181,182,183,184]
Ease of movement	Use of the different areas can be assessed through cameras, RTLS or GPS.	[185,186]
Presence of injuries	Detection through data images.	[182]
Absence of disease	The monitoring of animal behaviour, including their feeding, resting, and movement patterns, and circadian rhythms, can provide valuable information regarding the presence or absence of diseases. Furthermore, the identification of coughing episodes through sound recording may serve as an indication of respiratory disorders. Based on IoT devices, it was reported that accuracy on cows disease detection was approximately 98%	[177,178,180,187,188,189]
Absence of pain induced by management procedures	Monitoring via facial screening during the treatments.	[190,191,192]
Expression of social behaviours	The functioning of the social group can be determined on the basis of the animals’ interactions, positions, and activity, and it can be detected via RTLS monitoring.	[185,193,194]
Expression of other behaviours	Farmed animals’ utilization of resources that enhance their welfare can be monitored. For instance, accelerometers on brushes can detect the usage of brushes by cattle, while an RFID detector or a real-time locating system (RTLS) can track the animals’ proximity to the brushes. Similarly, the use of outdoor areas by poultry can be monitored with infrared beams that detect when the birds pass through the passage to the outdoor area.	[187,195,196]

**Table 4 sensors-23-08429-t004:** Schematic representation of main diseases on livestock production and detection methods.

Disease	Detection
Mastitis	Electrical conductivity; L-lactate dehydrogenase; colour; somatic cell count; homogeneity [187]; volatilome analysis
Lameness	Kinetic and kinematic measurements via radar, accelerometers, cameras, and electromyography
Postpartum disease	Localization; body temperature monitoring; activity measurement; rumination monitoring; parturition
Coccidiosis	Volatilome analysis
African swine flu	Camera monitoring

## Data Availability

No new data were created.

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
