# Peer review of "Nanotechnology and E-Sensing for Food Chain Quality and Safety"

_sensors, 2023, doi:10.3390/s23208429_

Round 1
Reviewer 1 Report
The manuscript contains many fields, topic is interesting. However, this article is not a qualified review. The summary of techniques and principles in the manuscript is so simple, and many paragraphs are only a simple list of some preliminary information. Because there are so many fields described, the manuscript contains a large number of useless briefs, such as the classification of olive oil, the production of beer and so on. Some parts even look like the instructions for the instrument. In a word, due to the lack of summary and induction of specific scientific research content in the manuscript, makes it only a popular science article rather than a qualified review. Authors are advised to narrow the scope of the content and carefully disaggregate the results contained in the latest high-quality research in these fields, instead of putting a lot of content together.
The language of the manuscript can be easily understood.
Author Response
Reviewer 1
The manuscript contains many fields, topic is interesting. However, this article is not a qualified review. The summary of techniques and principles in the manuscript is so simple, and many paragraphs are only a simple list of some preliminary information. Because there are so many fields described, the manuscript contains a large number of useless briefs, such as the classification of olive oil, the production of beer and so on. Some parts even look like the instructions for the instrument. In a word, due to the lack of summary and induction of specific scientific research content in the manuscript, makes it only a popular science article rather than a qualified review. Authors are advised to narrow the scope of the content and carefully disaggregate the results contained in the latest high-quality research in these fields, instead of putting a lot of content together.
Thank you for your valuable indication, your comments have allowed us to remarkably improve the quality of our manuscript. As suggested, scientific research has been added to better describe the role of nanotechnology in agro-food sector, the tools used and the results obtained.
Changes have been reported in section 2 - 3.1 – 4 - 5.2:
- Section 2, page 3, lines 108-124
- Section 3.1.1, page 10 lines 386-477
- Section 3.1.3, page 13, lines 582-617
- Section 3.1.4, page 15, lines 660-693
- Section 4, page 16, lines 747-753
- Entire section 5.2 “sensors for ammonia detection” page 17
Conversely, parts where there was an excessive description of the instrument (section 2.1.2, page 7, lines 246-265) and the parts too discursive were removed (olive oil classification in section 3.1.1, page 12, lines 480-506 and section 3.1.3, page 14, lines 619-633 concerning beer production).
The authors have deepened in the best way the topics covered by removing unnecessary and too discursive information. However, the scope of the content has not been restricted because the authors would like to show an overview of the application of nanotechnology in the agro-food field, from the beginning to the end of the food chain: from the field (e.g. for the production of cereals and thus the control of the presence of mycotoxins) or from farming (e.g. control of the absence of animal diseases) to the production of the commercial product as described in section 3.1.
Reviewer 2 Report
· Please check the manuscript more carefully and make the presentation more consistent. e.g., “figure 2” at line 137 should be “Figure 2”
· Check this manuscript to avoid English language errors carefully, especially grammar and spelling errors, typos, extra space, superscripts and subscripts, etc. e.g., line 241 244 247 255 497, and so on.
· Line 657 should be “Figure 9”
English is OK, no grammar mistakes mistakes.
Author Response
Reviewer 2
Please check the manuscript more carefully and make the presentation more consistent. e.g., “figure 2” at line 137 should be “Figure 2”. Check this manuscript to avoid English language errors carefully, especially grammar and spelling errors, typos, extra space, superscripts and subscripts, etc. e.g., line 241 244 247 255 497, and so on. Line 657 should be “Figure 9”.
Thank you for your valuable indication, your comments have allowed us to remarkably improve the quality of our manuscript. As suggested, authors took care of everything you asked for. The English was checked and corrected throughout the review.
Reviewer 3 Report
The paper provides an overview of the applications of E-sensing, Nanotechnology, Robotics, and Smart farming in the field of food and agricultural science. The author discusses the technical principles, developments, and applications in these fields, and also summarizes recent research findings. The article can be a valuable resource for beginners in related fields.
1.The term "Nanotechnology" mentioned in the introduction should provide a general background and introduce the entire article. It appears that the author overly emphasizes nanotechnology in the introduction. Nanotechnology and e-sensing should have equal focus, but the subsequent content clearly has a few times more information on e-sensing. It is advised to have a separate section for nanotechnology and reorganize the introduction.
2.The introduction introduces nanotechnology and e-sensing, but the headings "Robotics" and "Smart farming" in the fourth and fifth paragraphs can only be loosely associated with e-sensing. Only a part of these fields use e-sensing. If the discussion in robotics and smart farming only focuses on e-sensing in food chain quality and safety, it would align with the title's significance. However, the author introduces a lot of general content, and the focus of the whole article is not as described in the title.
3.After reading the entire article, I believe that the author's title should be something like "E-sensing, Nanotechnology, Robotics, and Smart farming for Agriculture." However, I do not recommend using such a title as it covers a broad range of fields. If the author wants to emphasize "food chain quality and safety," it would be better to cite literature from the field of food science and highlight the application of "quality and safety" in each technology.
4.The entire article reads like a table of contents, telling readers "if you want to learn about XX, read XX article." It only serves as a minimum passing grade for a review. As a good review, it should at least have a relationship of depth and breadth, emphasizing the iterative nature of the technologies (which is more excellent and innovative). It should also introduce relevant parameter conclusions to help readers determine whether to read the cited articles when they are interested.
The author's effort is commendable, but it seems that the author has not decided on the direction of their research. It is advised for the author to have clear research objectives and field and conduct in-depth research.
Author Response
Reviewer 3
The paper provides an overview of the applications of E-sensing, Nanotechnology, Robotics, and Smart farming in the field of food and agricultural science. The author discusses the technical principles, developments, and applications in these fields, and also summarizes recent research findings. The article can be a valuable resource for beginners in related fields.
1.The term "Nanotechnology" mentioned in the introduction should provide a general background and introduce the entire article. It appears that the author overly emphasizes nanotechnology in the introduction. Nanotechnology and e-sensing should have equal focus, but the subsequent content clearly has a few times more information on e-sensing. It is advised to have a separate section for nanotechnology and reorganize the introduction.
- Thank you for your valuable indication, your comments have allowed us to remarkably improve the quality of our manuscript. As suggested, the introduction has been completely reformulated: the concept of nanotechnology has been better explained. Nanotechnology used in the agri-food sector ranges from the use of e-sensing devices, to detect specific molecules, to nanoparticles to improve the characteristics of food packaging. Authors focus mainly on nanosensors because the focal point of this review is the use of e-sensing to improve food production and guarantee a high level of microbiological quality, safety, enzymes and oxidation during storage. Nanotechnology applications in the food industry can be utilized to support protected products and extend food shelf life.
2.The introduction introduces nanotechnology and e-sensing, but the headings "Robotics" and "Smart farming" in the fourth and fifth paragraphs can only be loosely associated with e-sensing. Only a part of these fields use e-sensing. If the discussion in robotics and smart farming only focuses on e-sensing in food chain quality and safety, it would align with the title's significance. However, the author introduces a lot of general content, and the focus of the whole article is not as described in the title.
- Thank you for your valuable indication, your comments have allowed us to remarkably improve the quality of our manuscript. As suggested, more information has been added in section 4, page 17, lines 747-753 regarding the use of e-sensing in agriculture. A section on monitoring the concentration of ammonia in the environment has also been added, which significantly affects the welfare of animals during breeding (section 5.2 "Sensors for ammonia detection" line 766). The authors have also deepened the topic of nanotechnology in the food sector and defined more comprehensively the purpose of the work in order to make the title (Nanotechnology and e-sensing for food chain quality and safety) as suitable as possible for the review (page 1). Nanotechnology covers a very wide field, from nanoparticles to the use of nanosensors for the detection of volatile molecules and this technology is fundamental for the food chain.
3.After reading the entire article, I believe that the author's title should be something like "E-sensing, Nanotechnology, Robotics, and Smart farming for Agriculture." However, I do not recommend using such a title as it covers a broad range of fields. If the author wants to emphasize "food chain quality and safety," it would be better to cite literature from the field of food science and highlight the application of "quality and safety" in each technology.
- Thank you for your valuable indication, your comments have allowed us to remarkably improve the quality of our manuscript. The authors emphasized the concept of quality and food safety by reporting articles and results in all section 3.1 (E-sensing and food). Nanotechnology can be used from the beginning to the end of the food chain in order to guarantee food safety and quality products as highlighted in lines 71-83.
In addition, the authors emphasize how nanosensors can overcome many of traditional technology's drawback. Tools are often compared and their advantages and disadvantages described:
- Section 2, page 3, lines 110-117
- Section 2, page 3, lines 135-140
- Section 2.1.1, page 4, lines 147-150
- Section 2.1.1, page 4, lines 162-168
- Section 3.1.1, page 10, lines 375-380
- Section 3.1.1, page 11, lines 436-440
- Section 3.1.3, page 14, lines 572-581
- Section 3.1.4, page 15, lines 660-692
- Section 4, page 17, lines 743-753
4.The entire article reads like a table of contents, telling readers "if you want to learn about XX, read XX article." It only serves as a minimum passing grade for a review. As a good review, it should at least have a relationship of depth and breadth, emphasizing the iterative nature of the technologies (which is more excellent and innovative). It should also introduce relevant parameter conclusions to help readers determine whether to read the cited articles when they are interested.
The author's effort is commendable, but it seems that the author has not decided on the direction of their research. It is advised for the author to have clear research objectives and field and conduct in-depth research.
- Thank you for your valuable indication, your comments have allowed us to remarkably improve the quality of our manuscript. As suggested, the authors explained more in detail the technologies used in the food field by reporting scientific articles and their results.
Changes have been reported in:
- Section 2, page 3, lines 111-117
- Section 2, page 3, lines 135-140
- Section 2.1.1, page 4, lines 147-150
- Section 2.1.1, page 4, lines 162-168
- Section 2.1.2, page 6, lines 219-224
- Section 3.1.1, page 10, lines 375-380
- Section 3.1.1, page 11, lines 436-440
- Section 3.1.1, page 12, lines 470-478
- Section 3.1.2, page 12, lines 523-555
- Section 3.1.4, page 13, lines 564-581
- Section 4, page 17, lines 743-746
Conversely, parts where there was an excessive description of the instrument (section 2.1.2, page 7, lines 246-265) and the parts too discursive were removed (olive oil classification in section 3.1.1, page 12, lines 480-506 and section 3.1.3, page 14, lines 619-633 concerning beer production).
The authors have deepened in the best way the topics covered by removing unnecessary and too discursive information. Especially regarding e-sensing and food, scientific research has been carried out and the concept of "quality and food safety" has been emphasized.
The authors would like to show an overview of the application of nanotechnology in the agro-food field, from the beginning to the end of the food chain: from the field (e.g. for the production of cereals and thus the control of the presence of mycotoxins) or from farming (e.g. control of the absence of animal diseases) to the production of the commercial product as described in section 3.1.
Reviewer 4 Report
An interesting article presenting the use of e-senses for process control of food production. I think many readers will be interested in this article. However, before it is made public, it is necessary to make appropriate corrections. Here are some of my observations and comments:
page , line 80-83, it is not emphasized what is the scientific novelty of this work. Since this is a review work, it should also be described what the authors critically refer to the state of knowledge in a given field. It needs to be supplemented.
page 3, line 106-109, The authors listed the applications of the electronic nose, and it must also be shown that the electronic nose has been used to assess the suitability for consumption of various food products, but also the electronic nose is great for assessing the authenticity of the product or product counterfeiting. Below are some works of the team of Prof. Namieśnika to the above-described problem:
doi: 10.1007/s00706-017-1969-x
page 3, line 112-117, it is also important to remember that a low price is also required from an ideal sensor, the cost of measurement is important and often determines the use of an electronic nose or GC-MS.
page 4, line 147-150, it should also be mentioned that in recent years the technology of bio-electronic nose, where the sensors are biosensors, has been developing very much. It's worth mentioning.
page 7, lines 253-259, it can also be added in this paragraph that the sensitivity of MOX sensors is influenced by: doping of the active surface with various metals and modulation of the sensor's working temperature.
page 12, line 485, I think it would be worth making a table summarizing the advantages and disadvantages of using e-noses and e-tongues in the diagnosis of food quality such as: coffee, beer, cereals, olive oil.
page 14, table 2, I think one of the controlled pollutants in livestock farming is ammonia, it would be useful to specify acceptable concentration levels for ammonia to achieve animal welfare.
page 18, line 671-675, currently odorless biogas plants are being implemented, I think that a few words should also be written on this subject, the more so that the condition of odorlessness is met, this pollution must be monitored.
page 19, line 707 I miss a chapter on the prospects for the development of sensors, electronic noses or electronic languages that would be used in the analysis of food from production to processing to ready consumption.
Author Response
Reviewer 4
An interesting article presenting the use of e-senses for process control of food production. I think many readers will be interested in this article. However, before it is made public, it is necessary to make appropriate corrections. Here are some of my observations and comments:
Page 2, line 80-83, it is not emphasized what is the scientific novelty of this work. Since this is a review work, it should also be described what the authors critically refer to the state of knowledge in a given field. It needs to be supplemented.
Thank you for your valuable indication, your comments have allowed us to remarkably improve the quality of our manuscript. As requested, the authors have explained in the best way the novelty of this work, emphasizing how nanotechnology can be of fundamental importance in the agro-food sector (section 1, lines 36-82).
Page 3, line 106-109, The authors listed the applications of the electronic nose, and it must also be shown that the electronic nose has been used to assess the suitability for consumption of various food products, but also the electronic nose is great for assessing the authenticity of the product or product counterfeiting. Below are some works of the team of Prof. Namieśnika to the above-described problem:
doi: 10.1007/s00706-017-1969-x
Thank you for your valuable indication, your comments have allowed us to remarkably improve the quality of our manuscript. As requested, parts relating to the use of the electronic nose have been added to assess the suitability for consumption of various foodstuffs:
- Section 2, page 3, lines 108-123 (work of the team of Prof. Namieśnika)
- Section 3.1.1, page 10, lines 386-478
- Section 3.1.3, page 13, lines 564-602
- Section 3.1.4, page 15, lines 660-593
Page 3, line 112-117, it is also important to remember that a low price is also required from an ideal sensor, the cost of measurement is important and often determines the use of an electronic nose or GC-MS.
Thank you for your valuable indication, your comments have allowed us to remarkably improve the quality of our manuscript. As suggest, authors highlight that MOX sensors are not only economical but also simple and flexible (lines 780-782). Also the following sections highlight the economic advantages of sensors compared of other analysis tools:
- Section 3.1.1, page 10, lines 375-380
- Section 3.1.1, page 11, lines 436-438
- Section 3.1.2, page 13, lines 537-539
- Section 3.1.2, page 13, lines 547-555
Page 4, line 147-150, it should also be mentioned that in recent years the technology of bio-electronic nose, where the sensors are biosensors, has been developing very much. It is worth mentioning.
Thank you for your valuable indication, your comments have allowed us to remarkably improve the quality of our manuscript. The authors mentioned biosensors citing a scientific article regarding the detection of mycotoxins. In section 3.1.4 lines 660-693 authors explained biosensor’s function.
Page 7, lines 253-259, it can also be added in this paragraph that the sensitivity of MOX sensors is influenced by: doping of the active surface with various metals and modulation of the sensor's working temperature.
Thank you for your valuable indication, your comments have allowed us to remarkably improve the quality of our manuscript. As suggested, authors added the factors that influenced the sensitivity of MOX sensors (page 7 lines 278-282).
Page 12, line 485, I think it would be worth making a table summarizing the advantages and disadvantages of using e-noses and e-tongues in the diagnosis of food quality such as: coffee, beer, cereals, olive oil.
Thank you for your valuable indication, your comments have allowed us to remarkably improve the quality of our manuscript. As suggested, the advantages and disadvantages of e-sensing have been added in the various paragraphs. Authors preferred to describe them through examples, reporting specific scientific articles.
The implementations are reported in:
- Section 2, page 3, lines 111-117
- Section 2, page 3, lines 135-140
- Section 2.1.1, page 4, lines 147-150
- Section 2.1.1, page 4, lines 162-168
- Section 2.1.2, page 6, lines 219-224
- Section 3.1.1, page 10, lines 375-380
- Section 3.1.1, page 11, lines 436-440
- Section 3.1.1, page 12, lines 470-478
- Section 3.1.2, page 13, lines 523-555
- Section 3.1.3, page 13, lines 564-581
- Section 4, page 17, lines 743-753
Page 14, table 2, I think one of the controlled pollutants in livestock farming is ammonia, it would be useful to specify acceptable concentration levels for ammonia to achieve animal welfare.
Thank you for your valuable indication, your comments have allowed us to remarkably improve the quality of our manuscript. As suggested, a section on ammonia has been added: “Sensors for ammonia detection” page 17 line 766.
Page 18, line 671-675, currently odorless biogas plants are being implemented, I think that a few words should also be written on this subject, the more so that the condition of odorlessness is met, this pollution must be monitored.
Thank you for your valuable indication, your comments have allowed us to remarkably improve the quality of our manuscript. As suggested, authors made a comment about odorless pollution (page 20 lines 884-887).
Page 19, line 707 I miss a chapter on the prospects for the development of sensors, electronic noses or electronic languages that would be used in the analysis of food from production to processing to ready consumption.
Thank you for your valuable indication, your comments have allowed us to remarkably improve the quality of our manuscript. No chapter has been made of the analysis of food from production to processing to ready consumption because the aim of the review is to explain how nanotechnology and therefore e-sensing are gaining ground in the agri-food sector from the beginning to the end of the food chain in order to ensure maximum food safety to consumers. For this reason, the topics have been treated in detail in the various chapters, starting from breeding and agriculture up to the final product:
- Animal welfare farming and the production of quality products (section 5 smart farming)
- Agriculture for the detection of potential plant diseases (section 4. Robotics applications and smart irrigation in agriculture and section 3.1.4 E-sensing and cereals)
- Production of foodstuffs such as oil, beer and coffee (section 3.1.1, 3.1.2, 3.13)
- The progress of food packaging with the use of inorganic nanoparticles (lines 53-66)
Nevertheless, authors talk about future works in the conclusion (lines 977-987). Comments highlighted in red have been added.
Reviewer 5 Report
The article "Nanotechnology and e-sensing for food chain quality and safety" is devoted to chemical gas sensing, which is an important and applied area nowadays. The use of E-nose technology to determine the quality or type of food products is an actual and modern scientific task.
It can be seen that the authors have done a great job in collecting the material (213 references), which is a great value for the review. Nevertheless, in my opinion, the article in this form cannot be published. I would recommend that it should be substantially reworked and republished.
The paper contains only 9 figures, which are (to put it mildly) uninformative and it is not clear where they came from: there are no references). The narrative style used in the paper is more suitable for writing popular science articles rather than reviews in peer-reviewed scientific journals. The paper contains the most trivial data, very few specifics and real facts. For example, response values or specific gases that would be markers of food quality. The entire 3rd section boils down to listing objects that can be detected with gas sensors without giving specific data. It seems to me that all data should be summarised in tables with parameters.
As a recommendation, I would advise the authors to look at similar reviews on gas sensing in peer-reviewed journals to see again what a review should look like, in a ranked journal.
Author Response
Reviewer 5
The article "Nanotechnology and e-sensing for food chain quality and safety" is devoted to chemical gas sensing, which is an important and applied area nowadays. The use of E-nose technology to determine the quality or type of food products is an actual and modern scientific task. It can be seen that the authors have done a great job in collecting the material (213 references), which is a great value for the review. Nevertheless, in my opinion, the article in this form cannot be published. I would recommend that it should be substantially reworked and republished. The paper contains only 9 figures, which are (to put it mildly) uninformative and it is not clear where they came from: there are no references). The narrative style used in the paper is more suitable for writing popular science articles rather than reviews in peer-reviewed scientific journals. The paper contains the most trivial data, very few specifics and real facts. For example, response values or specific gases that would be markers of food quality. The entire 3rd section boils down to listing objects that can be detected with gas sensors without giving specific data. It seems to me that all data should be summarised in tables with parameters. As a recommendation, I would advise the authors to look at similar reviews on gas sensing in peer-reviewed journals to see again what a review should look like, in a ranked journal.
Thank you for your valuable indication, your comments have allowed us to remarkably improve the quality of our manuscript. Regarding the images, they have no reference because they were provided by the authors. The aim of the review is to explain how nanotechnology and therefore e-sensing are gaining ground in the agri-food sector from the beginning to the end of the food chain in order to ensure maximum food safety to consumers. However, numerous changes have been made to the review, adding scientific articles and results obtained using nanotechnology:
- Section 2, page 3, lines 108-124
- Section 3.1.1, page 12, lines 386-477
- Section 3.1.3 page 13, lines 564-593
- Section 3.1.4, page 15, lines 660-693
- Section 4, page 17, lines 743-753
- Section 5.1.2, page 18, lines 852-859
Round 2
Reviewer 1 Report
Accept in present form
Author Response
Thank you for your valuable indication.
Reviewer 4 Report
I accept the answers given to me by the authors. Secondly, significant corrections have been made to the new manuscript. I think the article will arouse interest among readers. I recommend it for further stages of evaluation.
I think it's worth using a language editor to remove minor errors.
Author Response
Thank you for your valuable indication.
Reviewer 5 Report
The authors have done a lot of work to edit the review and it looks much better now. Nevertheless, I would recommend to make a common table, where data on the use of different gas analysing systems for food quality detection would be collected. The table should contain the following data: type of sensing element (specific MOS or others), number of receptor layers, detection operating temperature, gas analysed, response value, food product, reference. In my opinion the creation of such a table will be extremely useful for future readers.
Author Response
The authors have done a lot of work to edit the review and it looks much better now. Nevertheless, I would recommend to make a common table, where data on the use of different gas analysing systems for food quality detection would be collected. The table should contain the following data: type of sensing element (specific MOS or others), number of receptor layers, detection operating temperature, gas analysed, response value, food product, reference. In my opinion the creation of such a table will be extremely useful for future readers.
Thank you for your valuable indication, as suggested tables have been added:
- The first table is on line 621. Here the authors described the sensors and the sensitive elements used for the study of food (olive oil, coffee and beer).
- The second table is in line 702. Here the authors described some techniques that can be used to identify some of the mycotoxins in cereals.